# Synthesizing Navigation Abstractions for Planning with Portable Manipulation Skills

**Eric Rosen**[*]    **Steve James**[†]    **Sergio Orozco**[*]    **Vedant Gupta**[*]    **Max Merlin**[*]

**Stefanie Tellex**[*]                  **George Konidaris**[*]

**Abstract:** We address the problem of efficiently learning high-level abstractions for task-level robot planning. Existing approaches require large amounts of data and fail to generalize learned abstractions to new environments. To address this, we propose to exploit the independence between spatial and non-spatial state variables in the preconditions of manipulation and navigation skills, mirroring the manipulation-navigation split in robotics research. Given a collection of portable manipulation abstractions (i.e., object-centric manipulation skills paired with matching symbolic representations), we derive an algorithm to automatically generate navigation abstractions that support mobile manipulation planning in a novel environment. We apply our approach to simulated data in AI2Thor and on real robot hardware with a coffee preparation task, efficiently generating plannable representations for mobile manipulators in just a few minutes of robot time, significantly outperforming state-of-the-art baselines.

**Keywords:** Learning Abstractions, Mobile Manipulation

## 1  Introduction

Planning for mobile manipulation is difficult because of its long-horizon nature. There are two approaches to addressing this difficulty: subtask decomposition and structural decomposition. The former approach decomposes the problem into smaller subtasks (e.g: hierarchical planning [1, 2]), and leverages abstractions in two forms: action abstractions, also called *skills*, which package motor behaviors into a single invokable action, and perceptual abstractions, typically represented as grounded *symbols*, which compactly represent the relevant aspects of task state. Learned abstractions can address complex planning problems [3], but existing approaches are sample inefficient because they do not exploit structure present in the robot and the world. The second approach—structural decomposition—aims to design algorithms that do just that. Navigation stacks typically focus on building maps and localizing a robot in a map [4, 5], and using those maps to navigate to a goal via path planning [6]. Research in robotic manipulation structures the task of effectively interacting with objects [7] into component algorithms such as object recognition [8], interactive perception [9], grasp synthesis [10], kinematic motion planning [11], and learning for manipulation [12]. This approach can produce algorithms that generate useful behavior while avoiding learning entirely.

We propose to combine these two complementary approaches by exploiting structural assumptions to efficiently learn high-level abstractions. We begin by splitting abstractions to do with manipulation from those to do with navigation. Manipulation abstractions are expensive to learn but are typically object-centric and therefore portable, while navigation abstractions are *not* portable: how the robot should abstract its map pose and navigate between locations depends on the specifics of a single scene. Efficiently learning the navigation components of the abstraction, which must be relearned for each task, is thus critical. We therefore assume a given (pre-learned or hand-constructed)

---

[*]Brown University
[†]University of the Witwatersrand

7th Conference on Robot Learning (CoRL 2023), Atlanta, USA.

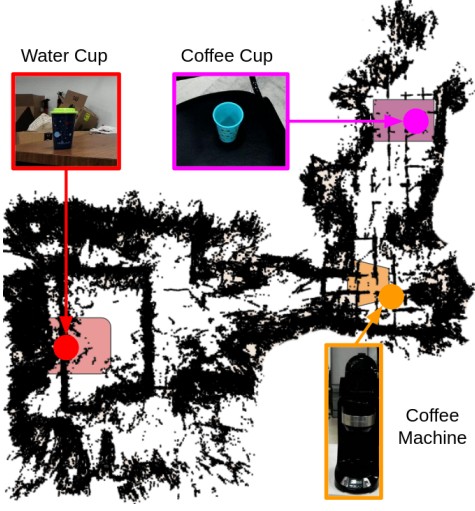

(a) An Action-Oriented Semantic Map for a coffee preparation task.

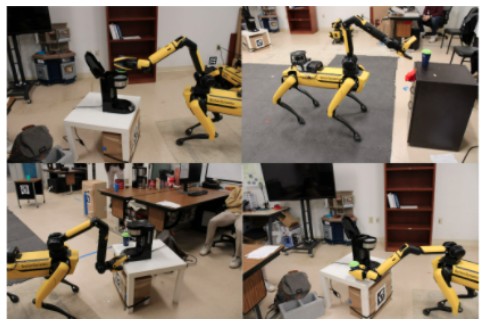

(b) Spot executing portable manipulation skills in coffee preparation task. Given a new environment with these objects, our approach efficiently constructs the navigation abstractions—both action and state—to support planning using these skills.

Figure 1: An AOSM for a coffee preparation task. (a) The underlying semantic map consists of a 3D point cloud of the scene (black points) along with the detected pose and attributes of objects. (b) Given a set of portable manipulation skills (start top left clockwise: pouring water, picking up a cup, placing a cup, and pushing a brewing button), an AOSM also includes a distribution over poses where the robot can execute each skill (visualized by colored areas in map (a)).

set of portable manipulation abstractions (both skills and symbols), and consider how to efficiently generate the navigation abstractions that support planning with them in a novel environment.

Our key insight is that spatial and non-spatial state variables typically contribute independently to whether a motor skill can be executed; and that under those conditions, a unique data structure—an Action-Oriented Semantic Map (AOSM) [13] (Figure 1a), which encodes the spatial locations from which manipulation skills can be executed—is necessary and sufficient to generate all the navigation abstractions required to support manipulation planning. We provide an algorithm to autonomously and efficiently construct an AOSM from a given set of manipulation skills using well-established mapping and path planning algorithms; a robot can thereby complete its abstract representation of a new task by constructing its navigation components in just a few minutes of robot time. We evaluate our approach in both simulation (using AI2Thor [14]) and on real robot hardware (a Boston Dynamics Spot). In simulation, our approach decreases the number of interactions required to learn navigation abstractions by an order of magnitude compared to the state of the art, and enables the robot to transfer learned symbols to new environments. On real robot hardware, our system generates a representation of a coffee-making task for two different kitchen environments in a few minutes.

## 2 Background

We adopt the Markov Decision Process (MDP) formalism for modeling agent decision-making. MDPs model the agent's environment as a tuple $(S, A, T, R, \gamma)$, where $S$ is the set of states, $A$ is the set of low-level actions, $T$ is a transition function describing the environment dynamics, $T(s'|s, a)$, $R$ is the reward function that expresses real-valued rewards, $R(s, a, s')$ and $\gamma \in [0, 1]$ is the discount factor. A policy $\pi(a|s)$ determines the probability of an action $a$ being executed in state $s$. Solving an MDP is equivalent to finding the policy that maximizes the sum of discounted future rewards: $J^\pi(s) = \mathbb{E}\left[\sum_{i=0}^{\infty} \gamma^i R(s_i, a_i, s_{i+1})\right]$.

**Abstract Representations** An abstract action set can reduce the problem diameter of solving an MDP by leaving lower-level controllers to resolve repeated subtasks. The options framework [15]

is the most popular abstract action framework. An *option* $o$ is a tuple with three components: an option policy $\pi_o$; an initiation set $I_o \subseteq S$ that identifies low-level states the option policy can be executed from; and a termination condition $\beta_o(s) \rightarrow [0, 1]$ that determines which states cease policy execution.

An advantage of using abstract actions (or motor skills) is that they need not necessarily be functions of the full problem state. For example, a motor skill for walking can just the robot's local perception, rather than an entire map. In such cases we model the option components as depending on some observation space $D$ obtained using sensor model $\phi(S) \rightarrow D$, and refer to the option as being portable since it can be reused in several places in a task, and in new tasks [16, 17].

We are interested in learning an abstract representation that facilitates planning. A probabilistic plan $p$ is sequence of (potentially abstract) actions to execute from states sampled from a distribution $Z$: $p_Z = \{o_1, ..., o_{p_n}\}$. A suitable representation for planning must enable the agent to correctly evaluate the probability of a plan. Konidaris et al. [3] proved that it is necessary and sufficient to learn when an option can be executed (known as the preconditions) and what the result of executing an option is (known as the image operator). Computing the image operator for arbitrary options is challenging; however, it is tractable for a subclass of *subgoal* options [18]. A subgoal option's resulting state distribution after executing the policy is independent of the starting state, so $Pr(s'|o, s) = Pr(s'|o)$. Therefore, computing the entire image operator can be substituted with representing the *effect* of executing the option (the distribution over states the agent will be in after executing the option), Effect($o$). An option that only modifies a subset of state variables—its *factors*– induces an abstract state space expressible using a classical planning representation like PDDL [3]. In this formulation, preconditions and effects can be represented by propositional symbols (which constitute an abstract state space), and actions are expressed as operators over those symbols. With an object-centric state-space, the learned symbols can instead be predicates parameterized by object types [19].

A two stage approach is used to learn a portable symbolic vocabulary and generate a forward model for a set of portable skills. First, symbols for the portable options are learned over the observation space $D$ in a training environment, then the portable options are partitioned in a test environment to make them subgoal in both $S$ and $D$. We defer the details of this process to James et al. [17, 20], and note that we use a similar approach for constructing our portable symbolic vocabulary. However, this formulation can take several hours and over a hundred skill executions to learn a representation for a simple task [3, 21, 22]; our main contribution is defining and leveraging the spatial independence property to make learning abstractions much more efficient.

**Related Work**   Our work focuses on learning state abstractions that enable long-horizon task planning by leveraging manipulation skills and semantic maps, similar to Task and Motion Planning (TAMP) frameworks. However, our work differs from TAMP based on the assumptions we make: Rather than use motion planning to generate manipulation behaviors, we treat manipulation skills as black-box skills that can be implemented with or without motion planning (e.g: learned motor policies [23]), and only require a model of the environment to support path planning for locomotion, which is readily accessible using off-the-shelf SLAM.

TAMP solutions integrate high-level task planning with low-level continuous motion planning to exploit a planning hierarchy where different specialized planning and learning algorithms can exploit the structure present at each level [24] and across modes [25]. However, whereas standard TAMP approaches assume access a given state abstraction is sound for a particular task [26, 24], we formalize an independence property between spatial and non-spatial state variables to more efficiently learn a sufficient representation for planning with given manipulation skills. Most similar to our work are TAMP approaches that leverage semantic maps for improving task and motion planning. Galindo et al. [27] investigate how semantic maps can act as a hybrid knowledge base for TAMP in the context of navigation. This work also uses a semantic map to improve task planning, but only extracts additional information from a semantic map, whereas we identify a specific augmentation to a semantic map that is provably sufficient for supporting manipulation planning. Our work is

also related to approaches that leverage Large Language Models (LLMs) for task planning. These approaches [28, 29] generally assume the existence of a preprocessed map that enables navigation to support manipulation. Our work here formalizes this data structure and lays the theoretical foundations for how this specific data structure can not just be used in task planning with LLMs, but also for learning symbols for task planning.

## 3 Exploiting Spatial Independence for Learning Abstractions

**Problem Definition**  We are interested in the problem of a robot that must navigate an environment and manipulate objects to achieve a goal. To this end, we represent the decision problem as an MDP, and factor the state $s \in S$ into the state of the robot $S_r$ and the state of the environment $S_e$: $S = S_r \times S_e$. Furthermore, the state of the environment can be factored into a discrete set of $q$ objects (or entities) the robot may manipulate, $S_\Omega = \Omega_1 \times ... \times \Omega_q$, and a map of the environment $m \in M$, $S_e = M \times S_\Omega$. This structured representation of the environment is often called a *semantic map* [14]. Since the robot and all of the objects exist in a physical space, they each have a pose in the map. Therefore, we factor the state of the robot $S_r$ into some pose $S_b$ in the map and any other information describing the state of the robot $S_r'$: $S_r = S_b \times S_r'$, and similarly for each object $\Omega_i \in \Omega$: $\Omega_i = \Omega_i^b \times \Omega_i'$. The task-specific semantic map defines a constraint function on the feasible poses of the robot, and can be used in conjunction with a path planner $N(s_b, s_b')$ to generate trajectories through the space of robot poses $S_b$ from a start state $s_b$ to a set of goal states $s_b'$ (i.e: locomote the robot around the scene).

Given the above setting, our problem is formalized follows. For a given set of portable manipulation options $O$ and a semantic map $S_e$, we must take plans that consist only of manipulation actions (called a manipulation-only plan $p_O = \{o_1, ..., o_{p_o}\}, \forall i \in \{1, ..., p_o\}, o_i \in O$, where $p_o$ is the length of the plan $p_O$), and learn a portable abstract representation that supports generating task-specific navigation behaviors based on $S_e$ that can be interleaved into the manipulation-only plan to make the probability of success non-zero. Note that even though the state space is fully observable, it crucially does not include information about what configurations in space afford manipulation, which is what our approach learns.

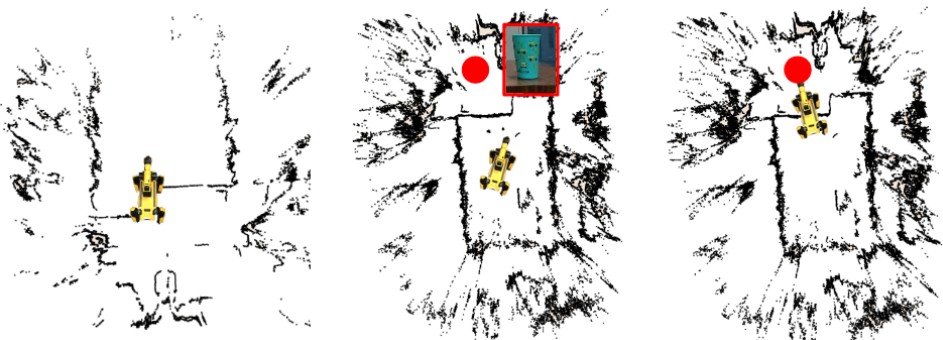

Figure 2: An example figure of a robot iteratively constructing an AOSM in a novel environment. (Left): The robot has a partial map of the environment and has not seen any objects. (Middle): The robot moves around to construct more of the map, and the vision model identifies a cup (position visualized as red circle). (Right): The robot uses a learned navigation symbol to sample a pose to pick the cup, and then navigates to that pose in order to execute the manipulation skill.

**Approach**  Our approach is based on autonomously constructing an Action-Oriented Semantic Map (AOSM) [13] and using it for task planning. Formally, an AOSM $(O, S_e, (V, E))$ is a data structure where $O$ is a set of $k$ portable manipulation options, $S_e$ is a semantic map, and $(V, E)$ is a topological graph. The topological graph $(V, E)$ is an undirected graph that contains $k$ nodes $V = \{v_1, ..., v_k\}$, where each node $v_j$ represents a region of configuration space for the base of the mobile manipulator (i.e: each node $v_j$ represents a set of poses in the semantic map). Node

$v_j$ corresponds to the set of poses in the semantic map that have a non-zero probability of being in the initiation set of option $o_j$. So, $v_j = \{p \in I_{o_j} | p \in m\}$. The node $v_j$ is also referred to as a navigation symbol $\sigma^{o_j}$ for the option $o_j$, since a symbol is a probabilistic binary classifier for testing membership of a set, and this symbol only depends on whether the robot's configuration is within a specific region of space that is relevant for navigation (discussed in more detail below). An edge $e = (v_a, v_b) \in E$ represents that a motion planner $N(v_a, v_b)$ can be used to successfully navigate from the set of poses in $v_a$ to the set of poses represented by $v_b$. AOSMs were introduced in Rosen et al. [13], where they were hand-crafted by a user. Here, we assume access to a set of portable manipulation skills $O$ and the semantic map $S_e$, and we provide a novel algorithm for learning the topological graph $(V, E)$ that consists of the navigation symbols and edge connectivity between them, which together define an AOSM.

When a robot has access to an AOSM, it can sample poses in the map that enable the robot to execute its manipulation skills (Figure 2). When the navigation symbols are learned in an object-centric spatial frame (i.e: the regions of space are in an object-centric frame instead of a map frame), they can be ported to new environments by grounding to global poses based on the known poses of the objects in the semantic map $S_e$. Once an AOSM has been constructed, given a manipulation-only plan $p_O = \{o_1, ..., o_{p_o}\}, \forall i \in \{1, ..., p_o\}, o_i \in O$, a starting base pose $S_b^0$, and a path planner $N(s_b, s_b')$, we can use the AOSM to sample poses from the navigation preconditions of each manipulation option $\{S_b^1, ..., S_b^{p_o}\}, \forall S_b^i \sim \sigma^{o_i}$, and leverage the the path planner to synthesize a sequence of locomotion path plans $p_N = \{n_1, ..., n_{p_o-1}\}, n_i \sim N(S_b^{i-1}, S_b^i)$ that can be interleaved into the manipulation plan $p_O$, $p_{O'} = \{o_1, n_1, o_2, n_2, ..., , o_{p_o-1}, n_{p_o-1}, o_{p_o}\}$. This augmented plan has the requisite additional actions required to make the manipulation-only plan feasible in the specific map the robot finds itself in. An AOSM can only can be used when it is possible to decompose initiation sets into navigation and manipulation preconditions. We now prove this assumes a crucial independence property of the factors of the initiation set, which we formally describe in the rest of this section.

First, note that we can define navigation symbol as a symbol $\sigma$ whose factors (the set of state variables the grounding classifier depends on) are the robot's mobile base state variables $S_b$, $\mathrm{Factors}(\sigma) = S_b$ (we call this type of factor a spatial factor). To determine whether a state variable is in the factor associated with the initiation set of a manipulation option (i.e: the state variable is a defining state variable for that set of states), we can use the notion of *projection*. The projection of a list of state variables $v$ out of a set of states $X$ is defined as $\mathrm{Proj}(X, v) = \{s | \exists x \in X, s[i] = x[i], \forall i \notin v\}$, which removes any restrictions on the values of the state variables $v$ for the states in $X$. If we project out a state variable from a set of states and it changes the set of states, we say that the state variable is a defining state variable for that set of states (since deciding whether a state is a member of X depends on a restricted value for $v$). If that set of states is the initiation set $I_o$ of an option $o$, then that collection of state variables is by definition the factors of $I_o$, $\mathrm{Factors}(I_o)$. In this case, the set of states describing the initiation set can be described by the intersection of independent state sets [3]. Formally, we say a factor $f_s$ is independent in the initiation set $I_o$ when: $I_o = \mathrm{Proj}(I_o, \mathrm{Factors}(I_o)/f_s) \cap \mathrm{Proj}(I_o, f_s)$. With this definition, we now define the spatial independence property:

**Definition 3.1** (Spatial Independence). The initiation set $I_o$ for an option $o$'s has the spatial independence property if:

$$I_o = \mathrm{Proj}(I_o, \mathrm{Factors}(I_o)/S_b) \cap \mathrm{Proj}(I_o, S_b). \tag{1}$$

Note that when learning a probabilistic symbolic representation, the sets are replaced with distributions and the intersection is replaced with multiplication, and therefore the independence property is defined exactly as conditional independence. When an option's initiation set has the spatial independence property, we can construct an independent symbol to represent $\mathrm{Proj}(I_o, \mathrm{Factors}(I_o)/S_b)$ which by definition is a navigation symbol since it it only depends on $S_b$. Intuitively, this projection represents the set of base locations the robot must be in order to successfully execute the option

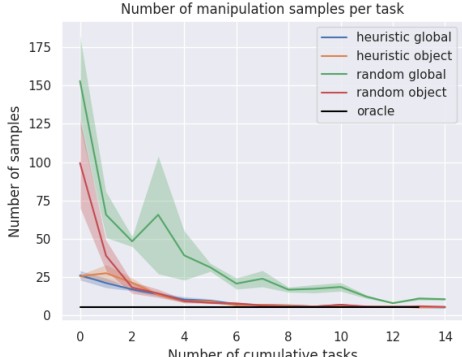
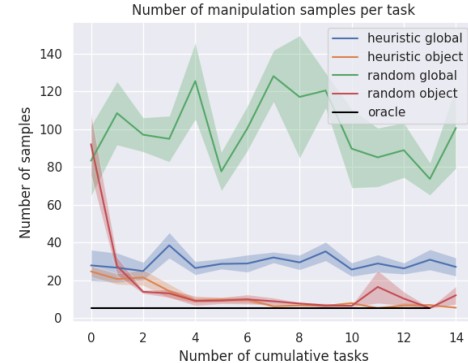

Figure 3: Results for our experiments on transferability of learning abstractions (left and right are single-scene setting/multi-scene setting respectively). We report the cumulative number of sampled locations that manipulation actions are attempted from against the average cumulative number of times the agent has successfully completed the plan (bars are standard error across 5 seeds.)

$o$ without regards to the state of the rest of the world. [3] Since an AOSM captures the navigation symbols, then when the spatial independence property holds for an option, an AOSM is a necessary and sufficient characterization of the spatial components of the initiation set. We leave details of the formal conditions under which we will resolve a manipulation option $o$ for some set of starting states $Z$ in the supplementary material. With an AOSM, given a manipulation-only plan, we can synthesize the requisite navigation actions to interleave into the plan and support execution. To evaluate the probability of the entire plan, we first learn a portable symbolic vocabulary similar to James et al. [17] (described in Section 2) but do not include spatial information about the objects or robot in the observations, and then separately learn navigation symbols using the spatial data in an object-centric frame. With the portable symbolic vocabulary, manipulation-plans can be generated, and with the addition of the navigation symbols grounded for a specific environment, we can evaluate the probability of a manipulation-only plan with navigation actions interleaved in.

## 4    Simulation and Hardware Experiments

We test the hypothesis that exploiting the spatial independence property of manipulation options increases sample efficiency and transferability of learned abstractions. First, we investigate the effect of leveraging the spatial independence assumption on the number of samples required to learn a useful set of abstractions for planning. Secondly, we evaluate the effectiveness of transferring abstractions from a training environment to a novel environment. Together, these experiments highlight how AOSMs can be used to efficiently learn and transfer abstractions with only a few number of interactions with the environment.

**Coffee Preparation Task**    We conduct both of our experiments in a simulated mobile manipulation domain, AI2Thor [14], using a coffee preparation task in 15 virtual kitchens. In this task the robot must navigate through a large simulated kitchen and manipulate objects; to successfully make coffee, it must pick up a cup, bring it to a coffee machine, turn on the coffee machine to make the beverage, and then pick up the prepared coffee mug. We assume the robot has access to a set of portable manipulation skills (**PickUp(Mug)**, **ToggleOn(CoffeeMachine)**, **PutIn(Mug,CoffeeMachine)**, **Make-Coffee(Mug,CoffeeMachine)**) that can be reused across different kitchen scenes, but that the agent must construct navigation abstractions for each different scene. AI2Thor provides semantic maps of each scene, which include a 2D occupancy grid of the environment, the number of objects in

---

[3]We note that this assumption may be violated in realistic domains (for example, the location of objects may constrain what locations the robot can execute a manipulation option from), but we later discuss how we can still use an AOSM to synthesize effective navigation abstractions even when this assumption is not met.

the environment, their object type and attributes, and their pose. We use 77 different objects, each characterized by a vector of length 108. We also include the 3D position and 1D yaw of the robot's base (4 additional state variables), resulting in a low-level observation vector of 8320 elements.

**Simulation Experiment: Spatial Independence for Learning Symbols**    In the first experiment, our goal is to evaluate how leveraging the spatial independence assumption affects the samples required to construct a symbolic vocabulary that supports planning. We therefore evaluate a state-of-the-art baseline [19] for learning symbols that does not incorporate the spatial independence assumption against an augmentation of the approach that does leverage the spatial independence assumption. We report performance as a function of the number of samples from the environment.

Part of the model learning process requires identifying which factors are independence since there is no a priori assumption about the structure of the initiation and effect sets of the skills. Partitioning is done via DBSCAN clustering [30], and the precondition classifiers are learned using a SVM [31] with an RBF kernel (hyperparameters are optimized using grid search. The effect density estimation is performed with a kernel density estimators [32, 33] with a Gaussian kernel, with a grid search over the bandwidth.

**Approaches**    We use a codebase for learning symbols [19] that is state-of-the-art but does not leverage any spatial independence assumptions as our baseline. More details on the algorithm can be found in [19], but in summary: the robot collects transition data in an environment by either randomly navigating to a pose or choosing manipulation skills to execute, and then uses this data to learn a model describing the preconditions and effects of the skills via a partitioning and clustering process. Part of the model learning process requires identifying which factors are independent since there is no a priori assumption about the structure of the initiation and effect sets of the skills. Details on the learning can be found in the supplementary material.

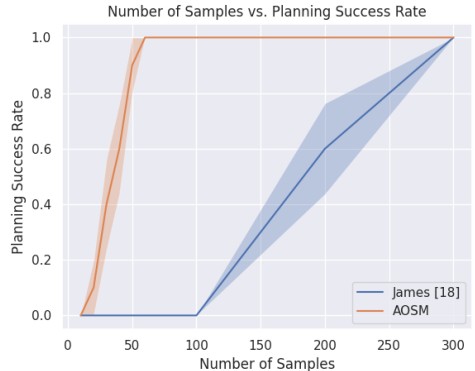

Figure 4: Learning symbols for the coffee preparation task, without the spatial independence assumption (James et al. [19]) and with the spatial independence assumption (AOSM). We report the number of sampled interactions with the environment against the planning success rate across 10 seeds.

**Metrics**    To evaluate the usefulness of the resulting abstractions, we use Fast Downward [34], an off-the-shelf symbolic planner, to plan using the resulting symbolic vocabulary. We then use a binary metric to determine how useful the representation is for planning: if the resulting plan accomplishes the goal, then the symbolic vocabulary is deemed successful. Otherwise, the symbolic vocabulary is deemed a failure. Our goal is to minimize the interactions required to learn a successful symbolic vocabulary for planning. We collect 1000 transitions with 10 different random seeds.

**Results**    The results of our experiment are in Figure 4. As the number of environmental samples increases, the success rate of planning with the symbols improves for both approaches, as expected. Learning with the spatial independence assumption, however, is able to learn a successful symbolic vocabulary with a nearly 100% planning success rate with about 50 samples, where as the baseline approach that does not leverage the spatial independence requires about 300 samples. This is due in part to the fact that, without leveraging the spatial independence assumption, the baseline requires more samples to learn to disentangle spatial information from non-spatial information, which is challenging since the spatial data is continuous. Our approach builds in the disentanglement between the spatial and non-spatial data, easing learning. These results demonstrate that our approach—

which structures in the independence assumption—is more sample efficient than state-of-the-art approaches to learning abstractions. Examples of the learned symbolic vocabulary are in Figure 5.

```
(:action pick-up-mug-from-table
        :parameters (?r - robot ?m - mug)
        :precondition (and (on-table ?m) (at-table ?r))
        :effect (and (not (on-table ?m)) (on-robot-hand ?m))
)
```

```
(:action PutMugInMachine-partition-0-8
        :parameters (?a - CounterTop ?b - CoffeeMachine ?c - Mug)
        :precondition (and (notfailed) (CounterEmpty2 ?a)
               (MachineOnEmpty ?b) (MugFilledHeld ?c) (AtMachine ?b))
        :effect (and (MugFilledInMachine ?c) (MachineOnHoldingMug ?b)
               (CounterHoldingMug ?a) (not (CounterEmpty2 ?a))
               (not (MugFilledHeld ?c)) (not (MachineOnEmpty ?b)))
```

Figure 5: Example operators for two manipulation skills with the navigation symbols injected into the preconditions (red highlight). (Left): A learned operator for the **PickUp(Mug)** skill in AI2Thor. Symbols are renamed manually to provide human interpretability (Right): A hand-specified operator for the **PutIn(Mug,CoffeeMachine)** skill in the Spot experiment.

**Simulation Experiment: Transfer of Learned Abstractions**   In the second set of experiments, we evaluate how AOSMs help transfer learned abstractions to novel environments. We provided a manipulation-only plan that prepares coffee, and the robot constructs the navigation symbols that enable it to generate supporting navigation behaviors. There are two important design choices when learning navigation symbols that can be chosen independently of each other: 1) which spatial frame are the navigation symbols learned in, and 2) what proposal distribution is used for rejection sampling. We evaluate different choices of these design choices in two settings: one where the robot learns symbols in a single scene, and one where it must learn symbols across different scenes (i.e: transfer is necessary). For each task execution in a scene, we report the cumulative total number of manipulation skills the robot executed, until the plan succeeded. Our results can be see in Figure 3, and full details of our experiments can be found in the Supplementary material. The main takeaway is that learning symbols in an object-centric frame is important for transferability.

**Robot Hardware Demonstration**   We demonstrate the effectiveness of AOSMs by executing a coffee preparation task on a Boston Dynamic Spot platform (Figure 1b).  In this task, the robot must gather coffee grinds and water, pour them both into a coffee maker, close the lid of the coffee maker, and push a button to turn it on.  We supply the robot with a set of portable manipulation skills **PickUp(CoffeeGrinds), PickUp(WaterCup), Place(CoffeeGrinds), Place(WaterCup), Pour(WaterCup), Pour(CoffeeGrinds), CloseLid(CoffeeMachine)** and **PushButton(CoffeeMachine)**, whose implementation on the robot can be seen in Figure 1b.  The objects are scattered around the room, requiring the robot robot to navigate the environment correctly to successfully execute the manipulation skills. Images and full details on the robot hardware demonstration and evaluation can be found in Supplementary material.

## 5   Limitations

While our approach leverages spatial structure to make learning abstractions for mobile manipulators more efficient, several of the input assumptions limit generality. Namely, our approach assumes a fully observable environment so the semantic map must be created before learning occurs. Future work will investigate learning in partially-observable environments, handling skill repertoires that are continuously parameterized, and operating in highly-dynamic and unstructured environments like outdoors.

## 6   Conclusion

We have introduced the spatial indepence property, and proven how it can be used to more effeciently learn navigation abstractions by building an Action-Oriented Semantic Map. Once a robot has built an AOSM, it can find and execute long-horizon mobile manipulation plans; in our work, a real robot was able to construct the relevant navigation abstractions using just several minutes of data. Our results offer a promising path to enabling real robots to learn task-level abstractions in practical amounts of time, a capability critical for complex, goal-directed behavior.

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
