# OpenReview forum: "Synthesizing Navigation Abstractions for Planning with Portable Manipulation Skills"
_robot-learning.org/CoRL/2023/Conference — CoRL 2023 Poster_

### Official Review · Reviewer_VDot · 2023-07-15

**Confidence:** 4
**Originality:** Good
**Technical Quality:** Good
**Clarity Of Presentation:** Good
**Impact:** 3

**Recommendation:**

Weak Accept: I recommend accepting the paper, but will not argue for my recommendation if the majority of other reviewers have a different opinion.

**Review:**

- The Introduction and some related work are well-written however the background is too long and takes up too much (mostly unnecessary) space, in my opinion.
- The contribution and methodological details are not clearly written. I did carefully read the corresponding parts multiple times and many details are still not precise, mostly due to writing. Let me ask a couple of questions:
  - l146: How can manipulation-only plans be made without considering navigation? Maybe some objects are in places that are not reachable.
  - l149: "v_j represents a region of space in the scene": what is space?
  - l150: "v_j corresponds to the set of poses": does region correspond to a set of poses?
  - l152: "v_j is also referred to as a navigation symbol": how is a navigation symbol related to a region or set of poses?
  I am confident that the authors have clear responses to these questions. However, the paper should have been written in a form that is consistent, precise, and clear in the first place.
- The method details are not given. What exactly is learned? How exactly does learning take place? Which structures are being used in learning? What is the classifier? How is it used? Actually, the paper provides a theoretical framework, provides the motivation, and then does not provide any methodological detail.
- The experiments and the importance of experimental results are difficult to comprehend. Whether the same assumptions are used by the baseline algorithm is not clear. The exact experiment parameters and the results obtained in the real robot are not provided in the manuscript.

**Quality Of The Limitations Section:**

Limitations are addressed clearly

**Questions For Rebuttal:**

The approach is compared with a learning-based baseline whose assumptions are not given. Given full observability, which means the exact locations of the objects, their manipulation poses, and the exact layout of the environment are known, does it make sense to compare the method with a standard non-data-driven planning framework?

**Robotics Focus:**

Sufficient demonstration on hardware

**Summary Of Paper:**

This paper argues that viewing and representing manipulation and navigation operators/symbols independently and learning navigation symbols/operators. Assuming a fully observable and known world, with full-manipulation-related knowledge (manipulation poses, object properties, etc.) in the world, the robot first makes a manipulation-only plan with the objects and then learns navigation abstractions between manipulation actions, executing a plan at the end that is composed of interleaved manipulation and navigation actions. The paper argues that this approach generates navigation abstractions efficiently compared to an approach where manipulation and navigation are not independent. The approach is verified in a simulation environment and shown to be sample efficient compared to a baseline, and realized in a real robot (where whether navigation abstractions are learned are not is not clear).

**Summary Of Recommendation:**

The idea of separating manipulation and navigation abstractions is interesting. However, the paper is not well written, the method is not provided clearly and in detail, and the experimental results are difficult to interpret given methodological and baseline-related uncertainties. Therefore, I believe that the paper was not ready for submission in its current form and should be significantly worked on. In its current form, how it does make a significant contribution to the literature is not clear.

Update:  I increase my score as the details of the method are more clear, and the idea of separating navigation and manipulation in learning for planning is novel.

---

### Official Review · Reviewer_Q4vG · 2023-07-20

**Confidence:** 2
**Originality:** Good
**Technical Quality:** Very Good
**Clarity Of Presentation:** Very Good
**Impact:** 3

**Recommendation:**

Weak Accept: I recommend accepting the paper, but will not argue for my recommendation if the majority of other reviewers have a different opinion.

**Review:**

**Strengths**

* The paper hits upon an important facet of robot planning - choosing state appropriately, particularly to leverage the structural properties of the task at hand, can vastly speed up the search problem. In this case, the authors leverage the fact that most _common_ manipulation skills need only be specified by their object frame, thereby allowing plans to generalize across different environments if the navigation aspects are left to be learned.
* The theory behind the problem formulation is well reasoned and articulated
* The results are impressive

**Scope for Improvement**

* The authors make a strong, but not necessarily generalizable claim, that an action-oriented sematic map is necessary and sufficient for generating navigation abstractions for mobile manipulation (line 42). The sentiment is expressed elsewhere in the paper. However, this is misleading, as the actions of picking and placing, for which the AOSM is particularly suited, are not the full space of mobile manipulation tasks. Other tasks (task types) exist, such as pushing a cart, inspection (particularly by robots such as Astrobee), collaborative moves, etc. for which a different structural decompositions of state, and hence a different "map" might be required to learn the right missing abstractions (not just navigation). I suspect the problem formulation does transfer, but it would be good for the authors to address this fact with appropriate evidence (or walk back their claims).
* The set of candidate locations for a skill, and the learning of them is crucial to this paper's learning of navigation symbols; yet that is in the appendix. More to the point, Figure 4, is incomprehensible without the content in Appendix 3 (as the terms in the legend are not defined in the main paper). Perhaps the authors should consider moving some of the details of "abstract representation" or "abstract actions" into the appendix to ensure that the details of skill learning are in the main paper?
* Lately, there are a new class of planning papers, such as Ahn, Michael, et al. "Do as i can, not as i say: Grounding language in robotic affordances." arXiv preprint arXiv:2204.01691 (2022) or Huang, Wenlong, et al. "Inner monologue: Embodied reasoning through planning with language models." arXiv preprint arXiv:2207.05608 (2022), etc. which provide an alternative to TAMP approaches, particularly for task planning. I recommend that the authors address a comparison to these works as part of their Related Work section as well---particularly how would this work complement (I suspect by providing a grounding(s)?) that work, or surpass it (I suspect in sample efficiency?), etc.

**Quality Of The Limitations Section:**

Limitations are addressed clearly

**Questions For Rebuttal:**

As the paper currently stands, it is very specific to pick and place (and button-press) type tasks. I recommend that the authors consider a wider class of tasks and posit state decompositions for those classes to make the relevance of this paper more apparent.

I also condition my acceptance of this paper on the inclusion of more details on learning the skill models in the main body of the paper (particularly as the evaluation of the approach is tied to this aspect too).

**Robotics Focus:**

Sufficient demonstration on hardware

**Summary Of Paper:**

* The authors propose a semantic map, and related state representation paradigm to allow agent plans to only be specified in terms of manipulation actions---leaving the "implementation" detail of navigation actions out of the plan that is specified to the agent
* In order to do this, the authors assume that manipulation skills, encoded as options in an MDP, are (a) specified w.r.t the reference object frame, and (b) are grounded to feasible locations from which the skills could be performed
* The authors show that by focusing on the structural parts of the plan that need to be inferred in the moment, in this case, the set of navigation actions required to complete the full plan, and updating state representation to match accordingly, the robot is able to learn the full set of symbols to solve a given planning problem more quickly than if state representation was not appropriately updated.

**Summary Of Recommendation:**

Conditioned on the suggested edits above, I am happy to accept this paper into the technical program for the conference.

---

### Official Review · Reviewer_zYUj · 2023-07-20

**Confidence:** 4
**Originality:** Good
**Technical Quality:** Good
**Clarity Of Presentation:** Very Good
**Impact:** 3

**Recommendation:**

Weak Accept: I recommend accepting the paper, but will not argue for my recommendation if the majority of other reviewers have a different opinion.

**Review:**

I find the major contribution of the paper is the idea of having the LLM build navigation/task abstractions that support mobile manipulation.

**Quality Of The Limitations Section:**

Additional details required

**Questions For Rebuttal:**

Add tasks.

**Robotics Focus:**

Sufficient demonstration on hardware

**Summary Of Paper:**

The paper proposes an algorithm to generate navigation task abstractions that are useful for efficient and precise mobile manipulation.

Shows results in a coffee making task on robot.

**Summary Of Recommendation:**

Weak accept.

---

### Official Review · Reviewer_yMcH · 2023-07-23

**Confidence:** 3
**Originality:** Good
**Technical Quality:** Good
**Clarity Of Presentation:** Very Good
**Impact:** 3

**Recommendation:**

Weak Accept: I recommend accepting the paper, but will not argue for my recommendation if the majority of other reviewers have a different opinion.

**Review:**

Strengths:

- The paper is generally well presented building on previously proposed formulations.
- The choice of experiments highlight the advantages of the proposed method and answer some logical scientific questions about the method.

Weaknesses:

- The authors do not motivate the need for the proposed method adequately in the abstract and the introduction.
- The proposed approach appears to be incremental and of limited value.
- There is a lack of exploration of the limitations that address scaling up the approach to more complicated real-world environments where there are more options, the environment is cluttered and there could be more than 1 task.

**Quality Of The Limitations Section:**

Additional details required

**Questions For Rebuttal:**

-Is the AOSM task specific? For each new task would there need to be a new semantic map?
-There are some points of ambiguity in the text. For example line 23 the authors state "the structure" although it is not clear what structure they are referring to. Same with "subgoal condition" line 89. Lines 131-133
-The order of figures referenced in text is not the same as the order they appear on-page.
-In line 301, the authors state that the objects are "scattered around various rooms", were there multiple rooms? The images and video appear to show a single room.
-There is a sentence that carries across 6 lines from line 83. It is a bit of a mouthful and hard to follow to the end being so long.

**Robotics Focus:**

Sufficient demonstration on hardware

**Summary Of Paper:**

The authors propose to improve the efficiency of learning navigation abstractions that tie into a predefined manipulation plan to complete a mobile manipulation task. The key improvement proposed comes from exploiting structure within a mobile manipulation task that splits the task into manipulation and navigation components. The approach relies on an Action-Orientated Semantic Map with the authors proposing a method for learning the topological graph of nodes and edges with the nodes corresponding to the feasible initiation set of manipulation options. The authors show the success of their approach in simulated experiments and on real hardware for a single coffee preparation task.

**Summary Of Recommendation:**

The authors proposed method is fairly well presented although lacking a strong motivation. The method appears technically sound with results backing the claims made. Results are presented from simulated and real-world experiments. The method is incremental with the value and contribution appearing to be limited.

---

### Author Response · Authors · 2023-08-15
**Updated manuscript**

We would like to thank all the reviewers for their constructive feedback, which we have included into the updated manuscript (attached to our rebuttal comment to Reviewer Q4vG). We appreciate Reviewer VDot stating they would “increase [their] score assuming those details (which are very important in my opinion) are added to the main manuscript.”, and Reviewer Q4vG engaging with our rebuttal and saying that if we are“ able to provide an updated version of the manuscript that we reviewers can evaluate, then I would be more likely to update my rating.” We have made the relevant changes and hope we have properly addressed the reviewer's comments. We state here for ease of reference what relevant changes were made, and are happy to make further changes if there is any other improvements the reviewers would like to see:

Reviewer Q4vG and VDot:
- We have moved the details on how the skills are learned from the Supplementary material to Section 4 inside the experiments section.

 Reviewer Q4vG:
- We have updated the Related Work section to include how this work relates to LLMs for task planning.

Reviewer VDot
- We have clarified in the problem statement and methodology about how all the v_j interpretations are equivalent to each other to make the learning objective and motivation clearer.

Reviewer  yMcH
- We have shortened the background to include more emphasis that the motivation of our work is to make learning abstractions for mobile manipulation more sample efficient over state-of-the-art approaches.
- We have clarified there was only one room in the hardware experiment (and multiple rooms in the simulation), and emphasized that the Action-Oriented Semantic Map is not task-specific, but it is environment specific (similar to a standard semantic map).

---

### Decision · Program_Chairs · 2023-08-30

**Decision:**

Accept (Poster)

**Comment:**

The paper introduces an approach to leverage manipulation abstractions to learn navigation abstractions. This choice of state abstraction enables an efficient representation for mobile manipulation planning.

Through the rebuttal the authors have engaged with reviewers and improved the paper, and the three reviewers recommend accept, which I concur with.

The paper could be further improved by (1) creating a full and more detailed algorithm figure and by (2) discussing related work on semantic representations for object search, e.g., https://arxiv.org/abs/2203.10421, https://arxiv.org/pdf/2209.09874.pdf, and https://proceedings.neurips.cc/paper/2020/file/2c75cf2681788adaca63aa95ae028b22-Paper.pdf. Though the paper focuses on a task and motion planning setting, I believe such works are quite relevant to the final state.